# CAUSAL REASONING FOR CONTROLLABLE GENERATIVE MODELING

## ABSTRACT

Deep generative models excel at generating complex, high-dimensional data, often exhibiting impressive generalization beyond the training distribution. However, many such models in use today are black-boxes trained on large unlabelled datasets with statistical objectives and lack an interpretable understanding of the latent space required for controlling the generative process. We propose CAGE, a framework for controllable generation in latent variable models based on casual reasoning. Given a pair of attributes, CAGE infers the cause-effect relationships between these attributes by estimating unit-level causal effects. We design a geometric procedure for estimating these effects that applies broadly to any latent variable model. Thereafter, we use the inferred cause-effect relationships to design a novel strategy for controllable generation based on counterfactual sampling. Through a series of large-scale human evaluations, we demonstrate that generating counterfactual samples which leverage the underlying causal relationships inferred via CAGE leads to subjectively more realistic images.

## 1 INTRODUCTION

Data generation using generative models is one of the fastest growing usecases of machine learning, with success stories across the artificial intelligence spectrum, from vision (Karras et al., 2020) and language (Brown et al., 2020) to scientific discovery (Wu et al., 2021) and sequential decision making (Chen et al., 2021). However, translating these successes into deployable applications requires an interpretable understanding of the generative process for the underlying black-box models—informing strategies for controllable generation in high-stake domains.

The application of causal reasoning is a natural framework to study and understand a generative model (Xu et al., 2020), and as a result is a crucial ingredient for effective controllability. Specifically, we are interested in causally informed controllable generation of pretrained deep latent variable generative models (DLVGMs), such as generative adversarial networks (GANs) Goodfellow et al. (2014) and variational autoencoders (VAEs) Kingma & Welling (2013). Prior work in controllable generation for DLVGMs can be categorized as either augmenting the loss function with disentanglement objectives, or post-hoc control approaches based on latent space explorations. The former class of methods e.g., Chen et al. (2016); Higgins et al. (2016) often introduce undesirable trade-offs with the standard generative modeling objectives (Locatello et al., 2019) and moreover, cannot be readily applied to the zoo of state-of-the-art pretrained models in use today.

We focus on post-hoc control of DLVGMs, where control is specified w.r.t. a small set of interpretable meta-attributes of interest, such as gender or hair color for human faces. Here, the predominant approach for controllable generation is to identify directions of variation in the latent space for each attribute and manipulate the latent code for an input image along these directions to achieve the desired control, e.g., Shen et al. (2020); Härkönen et al. (2020); Voynov & Babenko (2020); Khrulkov et al. (2021). While this procedure may work well in practice, it sidesteps a more fundamental question: what are the interactions between these attributes in the latent space of a deep generative model? Certainly, real world data generation processes have rich interplay—that goes beyond mere correlation—between such meta-attributes. As a result, in practical scenarios, we do not expect that the model perceives these meta-attributes as independent and hence, inferring the cause-effect relationships can enhance our understanding and controllability of DLVGMs.

We propose CAGE, a framework for inferring implicit cause-effect relationships in deep generative models. Our framework builds off the potential outcomes framework of the Neyman-Rubin causal model (Neyman, 1923; Rubin, 1974) and defines a new notion of *generative* average treatment effects (GATE). A fundamental problem of causal inference is that, by construction, we cannot observe the potential outcomes under all treatments (Holland, 1986) i.e., at any given time, any individual can be assigned only one treatment (a.k.a. the factual) but not both. However, when studying treatment effects for deep generative models, CAGE exploits the generative nature of such models to overcome this challenge and explicitly generate the counterfactual via a standard latent space manipulation strategy. Thereafter, we use an outcome attribute classifier to quantify the difference in outcomes for the factual and counterfactual generations and thus, estimate GATE. We refer to our overall framework for causal probing of deep generative models as CAGE.

We show that it is possible to further leverage the treatment effects estimated via CAGE to define a natural measure of causal direction over a pair of meta-attributes (e.g., gender and hair color) associated with a deep generative model. Unlike traditional causal discovery algorithms, CAGE infers causal structure over auxiliary metadata rather than over the observations themselves. Finally, we use these inferred causal directions for augmenting post-hoc strategies for controllable generation. Empirically, we show that knowledge of the causal directions inferred via CAGE significantly improves the generation quality over baseline controllable generation strategies on the CelebAHQ.

## 2 CAUSALLY AWARE CONTROLLABLE GENERATION

We approach the challenge of controllable generation from the perspective of causality and posit the existence of implicit causal structure within a deep generative model, $G$. We study this structure over meta-attributes, for example gender and hair color, with the underlying motivation manipulations which respect the implicit causal structure will yield more fine-grained control of $G$.

Formally, we assume full access to a pretrained DLVGM $G : \mathcal{Z} \rightarrow \mathcal{X}$. We further assume a finite dataset of annotated observations, $D = \bigcup_{i=1}^{n} \{\mathbf{x}_D(i), m_1(i), m_2(i)\}$. Here, the examples $\{\mathbf{x}_D(1) \ldots, \mathbf{x}_D(n)\}$ are drawn from the model's training distribution and we annotate each example $\mathbf{x}_D(i)$ with binary metadata for two variables $m_1(i), m_2(i) \in \{0, 1\}$. These two annotated variables can be drawn from a larger *unobserved* (w.r.t. $G$) causal process and apriori, we do not know which of the variables (if any) is cause or effect but seek to identify their relationship within $G$.

We note that any causal conclusions obtained under our proposed framework reflect implicit properties of a generator $G$, as opposed to properties of the data on which the generator was trained. That is to say, it is perfectly permitted, for $G$ to have learned a causal relationship that is inconsistent with the observational data used for training. As we shall see in §3, shedding light on the implicit causal relationship learned by $G$ is useful for controllable generation. As a result, this work—by design— is in sharp contrast with the majority of causal inference literature that focuses on identifying the generative mechanisms of a fixed dataset as opposed to our focus on the generative model itself.

Our framework proposes to extend the potential outcomes framework (see in §B.1 for a brief overview) to accommodate DLVGMs. Our approach is premised on computing a *generative* average treatment effect for a candidate causal attribute, $m_c$, on a given effect attribute, $m_e$, for a given generator, $G$. Unlike traditional approaches to treatment effect estimation, the key observation in CAGE is that we can simulate the counterfactuals under a given treatment.

### 2.1 OVERVIEW OF THE CAGE FRAMEWORK

In order to simulate the assignment of counterfactuals, we follow a three-step procedure.

**Step 1:** Our first step is to train a latent space classifier for the attributes using $D$. Here, we use the encoder to first project every input $\mathbf{x}_D(i)$ in $D$ to its latent encoding $\mathbf{z}_D(i) = \text{ENC}(\mathbf{x}_D(i))$.[1] Given the latent encodings $\mathbf{z}_D(i)$ and their annotations for $m_c$ and $m_e$ attributes, we train a (probabilistic) linear classifier, $\phi_c : \mathcal{Z} \rightarrow [0, 1]$, to discriminate between binary causal attribute values given latent representations. Concretely, by restricting ourselves to linear classifiers we obtain a unit hyperplane that encodes the classification boundary with $\mathbf{h}_c$ as the normal to the hyperplane for attribute $m_c$.

---

[1] For stochastic encoders such as in VAEs, we consider the mean of the encoding distribution.

**Step 2:** We use $G$ to create a dataset of annotated examples by sampling $k$ latent vectors $Z = \bigcup_{i=1}^{k}\{\mathbf{z}(i)\}$ from the prior of the generative model. Let $X = \bigcup_{i=1}^{k}\{\mathbf{x}(i) = G(\mathbf{z}(i))\}$ denote the corresponding generations. Since sampling latents from the prior generative model are typically inexpensive, our dataset size can be fairly large. For each generated example $\mathbf{x}(i) \in X$, we obtain its factual treatment by simply using the latent space classifier as $m_c(i) = \mathbb{1}[\phi_c(\mathbf{z}(i)) > 0]$.

**Step 3:** For a generated sample, $\mathbf{x}(i) \in X$, assume without loss of generality, that $m_c = 0$ is the factual treatment obtained via Step 2. We can derive analogous expressions for factual treatments $m_c = 1$, but we skip those for brevity. Finally, we define the counterfactual latent with respect to setting the treatment attribute $m_c = 1$ as using the **do** operator as: $\mathbf{z}(i; \mathbf{do}(m_c = 1)) = \mathbf{z}(i) + \alpha \mathbf{h}_c$, where $\alpha \in \mathbb{R}_+$ is a positive scalar that controls the extent to which we manipulate. Such a manipulation strategy corresponds to moving linearly along the hyperplane normal, $\mathbf{h}_c$, which encodes attribute $m_c$ for simulating the counterfactual. Such an approach is premised on the assumption that $\mathcal{Z}$ is linearly separable with respect to a semantically meaningful latent attribute, which was first observed and empirically validated for GANs (Denton et al., 2019). A counterfactual sample can subsequently be obtained by simply pushing forward the counterfactual latent through $G$:

$$\mathbf{x}(i; \mathbf{do}(m_c = 1)) = G(\mathbf{z}(i; \mathbf{do}(m_c = 1))). \tag{1}$$

Fig. 1 illustrates our counterfactual manipulation strategy.

### 2.1.1 QUANTIFYING TREATMENT EFFECTS VIA PROXY CLASSIFIERS

Equations (1) defines the geometric manipulations required to obtain interventional samples $\mathbf{x}(i; \mathbf{do}(m_c = 1))$ from a generative model, $G$. However, in order to estimate treatment effects, we require an estimate of the presence or absence of the effect attribute, $m_e$, for the counterfactuals. To this end, we propose the use of probabilistic binary classifiers, $\psi_e : \mathcal{X} \to [0, 1]$, trained to detect the presence of an effect variable i.e., classify counterfactuals generated via Equation (1). In this manner, we can then quantify the effect of intervening on the treatment attribute $m_c$ for an individual $\mathbf{x}(i)$ as:

$$m_e(i; \mathbf{do}(m_c = 1)) = \psi_e\left(\mathbf{x}(i; \mathbf{do}(m_c = 1))\right).$$

We subsequently the *generative* average treatment effect (GATE) of $m_c$ on $m_e$ as:

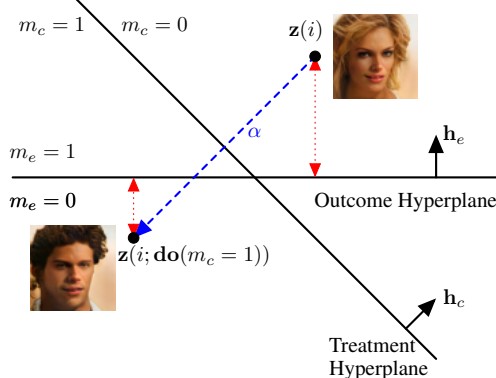

Figure 1: Counterfactual Manipulation in the Latent Space of Generative Models in CAGE.

$$\tau_{\text{GATE}}(m_c \to m_e) = \mathbb{E}_i[m_e(i; \mathbf{do}(m_c = 1)) - m_e(i; \mathbf{do}(m_c = 0))] \tag{2}$$

As we can see from Equation (2), the correctness of our estimate relies on a number of assumptions. In addition to standard assumptions from causality (unconfoundedness, positivity, SUTVA), the quality of the generative model and classifiers play an important role. For the generative model, we designed our counterfactual manipulation scheme assuming that the latent space is linearly separable in the attributes of interest. Further, implicit in our framework is the assumption that the generative model can generalize outside the training set to include the support of the counterfactual distribution. On the use of classifiers, we need accurate and calibrated classifiers (ideally, Bayes optimal) for both manipulating latent vectors and quantifying treatment effects. While it is impossible to test the above assumptions on real-world distributions, there is significant empirical evidence in the last few years that suggest that modern deep generative models and classifiers can indeed satisfy the above requirements for many practical usecases (Denton et al., 2019; Brown et al., 2020).

**$\Delta\tau$ as a measure of causal direction.** Finally, we note that the preceding sections have focused on the challenge of quantifying if a variable, $m_c$, has a causal effect on second variable, $m_e$. This assumes knowledge of the causal ordering over variables and maybe considered a special case of the more general causal discovery problem. As such, we can extend our approach to define a measure of causal direction over any pair of variables by considering the difference in absolute generative ATE scores when either variable is considered as the treatment:

$$\Delta\tau = |\tau_{\text{GATE}}(m_c \to m_e)| - |\tau_{\text{GATE}}(m_e \to m_c)| \tag{3}$$

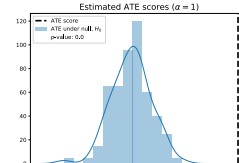 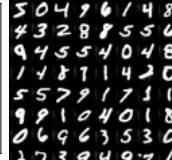 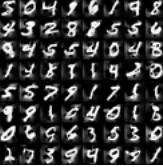 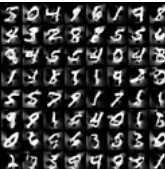

**Figure 2: Left-1**: A scatterplot of MoG with points colored by the value of causal attribute, $m_c$. **Left-2**: Histogram of GATE values under null hypothesis, and vertical line denoting estimated GATE. **Left-3** Generating counterfactual digits by reducing the thickness of each digit without changing the average intensity. **Left-4,5** Progressively decreasing digit thickness leads to background pixels being illuminated as the digits get thinner.

The $\Delta\tau$ score determines the magnitude of the difference in outcomes when either attribute is prescribed as the treatment. Thus $\Delta\tau > 0$ corroborates that the chosen causal ordering is indeed the one supported by $G$, while a $\Delta\tau < 0$ implies that the chosen causal ordering may in fact be reversed.

**A null distribution for treatment effects.** A further important consideration relates to how we might determine whether an estimated GATE is statistically significant. To address this, we can obtain an empirical sample of GATE scores under the null hypothesis where the attribute $m_c$ has no causal association with $m_e$. A simple manner through which this can be obtained is via randomization and permutation testing. We can randomly shuffle the values of a causal attribute, $m_c$, thereby removing any potential causal association. This corresponds to performing interventions over the latent space of $G$ which are effectively random projections. This process is repeated many times to obtain an empirical distribution for a GATE under the null hypothesis. Given an empirical distribution of GATEs under the null, we can obtain a $p$-value for our observed GATE.

# 3 EXPERIMENTS

We investigate the visual fidelity of counterfactual samples crafted by manipulating the causal variable or the effect variable as uncovered through CAGE in both synthetic datasets and CelebaHQ.

**Mixture of Gaussians**. We consider the following toy setup: data is generated according to two distinct mixture distributions as shown in the left panel of Figure 2 where each color denotes a mixture. We define the causal attribute, $m_c$, to be which mixture each sample is drawn from (i.e., from the mixture of 8 Gaussians, in blue, or the mixture of 3 Gaussians, in orange). We further define an effect variable, $m_e$, to be defined as one, if the mean of the $(x, y)$-coordinates is less than 2.5 and zero otherwise. In this manner, $m_c$ has a causal influence over $m_e$ by determining which mixture each sample is drawn from. We employ a Masked Autoregressive Flow (Papamakarios et al., 2017) as the deep generative model. The middle panel of Figure 2 visualises the distribution of $\tau_{\text{GATE}}$ under the null as well as the estimated GATE, $\tau_{\text{GATE}}(m_c \to m_e)$, which is significantly larger in magnitude while the right plots $\Delta\tau$ as a function of $\alpha$, positive values corroborate that $m_c \to m_e$.

**MorphoMNIST**. In our second synthetic experiment we evaluate the causal relationships learned by powerful deep normalizing flow models on a synthetic dataset based on MNIST dubbed MorphoM-NIST (Pawlowski et al., 2020). Here, the original MNIST digits are modified to respect a causal structure whereby the stroke thickness of the digits is a cause to the brightness (i.e. $T \to I$). Specifically, thicker digits are brighter while thinner digits are dimmer under the prescribed causal graph. We train a powerful normalizing flow in Mintnet (Song et al., 2019) and interrogate the direction of causality—if any—between thickness and intensity by first projecting all test set samples to their corresponding latent vectors. Figure 2 shows examples of counterfactuals, where we observe that the generated samples reduce the thickness while maintaining the average intensity. Specifically, the model generates digits that contain holes but to compensate for the drop in intensity the model instead increases the intensity of the background pixels. Such generations are in line with expectations as $\Delta\tau > 0$ (see §C.2) indicates that the model has learned that $T \to I$. As such manipulating the causal variable propagates influence to the effect variable yielding increased background intensity.

SAMPLE QUALITY OF COUNTERFACTUAL GENERATION

**Experimental Design**. For six pairs of metadata attributes we first look to infer the causal association between attributes using equation (3). The results are described in Appendix A and summarized

in the left column of Table 2. For each pair of attributes, we are therefore able to infer the implicit causal structure present within the generator, $G$ (see §C for additional experimental details).

Given the underlying causal structure, we consider two experimental paradigms: (1) same source and (2) same destination manipulation. In the same source setting, the starting point for generation is a source image (e.g., male with no mustache) which is then manipulated via changes to either the causal variable or the effect variable respectively. This produces two different images (e.g., female without a mustache and male with a mustache). In the same destination setting, we reverse the previous process by attempting to arrive at the same set of meta-attributes for two different real starting images after manipulation. For example, assume we wish to generate images of men without mustaches. Here, we may start from a female with no mustache and change its gender, or start from a male with a mustache and remove the mustache. For each attribute pair and both of the above settings, we generate 100 pairs of images which are evaluated using human annotators solicited through Amazon Mechanical Turk to rate images on the basis of their realism.

On the other hand, when doing conditional generation in tasks such as in-painting it is more desirable to modify the effect variable to produce higher quality visual samples.

We report the proportion of human annotators which preferred images generated by manipulating causal metadata variables in Table 1. In the same source setting, generating counterfactuals by modifying the causal variable leads to subjectively better visual samples as determined by human annotators in all but two settings (e.g., Gender → Bald/Blond hair, where the quality of samples indistinguishable). Conversely, in the same destination setting modifying the effect variable is preferable to the

Table 1: Counterfactual Sample Quality Results. Each cell represents the percentage of annotators that preferred the counterfactual generated by modifying the causal variable. $^*$ denotes statistical significance at level $\alpha = 0.05$, after Bonferroni correction.

| CAGE direction | Same Source ↑ | Same Dest. ↓ |
|---|---|---|
| Gender → Mustache | 231/300$^*$ | 51/300$^*$ |
| Gender → Bald | 156/300 | 138/300 |
| Gender → Goatee | 254/300$^*$ | 91/300$^*$ |
| Age ← Bald | 281/300$^*$ | 56/300$^*$ |
| Gender → Rosy Cheeks | 183/300$^*$ | 97/300$^*$ |
| Gender → Blond Hair | 157/300 | 126/300$^*$ |

causal one when evaluating for sample quality. These results highlight that depending on how $G$ is used downstream—e.g. applications such as data augmentation for rare classes, a higher degree of visual fidelity can be obtained through controllable generation using a causal attribute.

## 4 DISCUSSION

We proposed CAGE, a framework for inferring cause-effect relationships within the latent space of deep generative model via geometric manipulations for causally guided controllable generation. CAGE is well suited to a wide family of modern deep generative models trained on complex high-dimensional data and does not require any altering of the original training objective nor hard to obtain counterfactual data. Empirically, we have demonstrated that leveraging causal insights supplied by CAGE can lead to notable improvements in controllable generation, as verified by a large scale human evaluation study. In particular, our study revealed that higher quality samples are generated by controlling the causal variable in same source settings while manipulating the effect variable leads to a better fine-grained generation in same destination settings.

**Limitations and Future work**. Despite CAGE's broad applicability to deep generative models, one important limitation is the requirement of a latent space and as a result fully observed generative models (e.g., autoregressive modes) may need to adapt their internal representations for probing via CAGE. In addition, the notion of GATE in CAGE, while being novel and tested rigorously empirically, does not entirely follow the same analysis as ATE. We implicitly borrow some key assumptions regarding unconfoundedness and introduce new ones such as linear separability in semantic attributes of the latent space in designing our estimator. These assumptions may not always hold in practice and future work can investigate both theoretically and empirically if and when mitigation strategies are needed. Finally, as we consider only binary attributes, the extension of our framework to non-binary attributes and more than two variables is an interesting direction for future work.

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

# A    CAUSAL RELATIONSHIPS IN STYLEGAN2 OVER CELEBAHQ

We consider learning causal structure over six pairs of metadata attributes, as described in the table below. For each pair, $(m_1, m_2)$, the top row of figures visualizes sampleswhere $m_1$ is the treatment whilst the bottom utilizes $m_2$ as the treatment. We also visualize the estimated GATE $\delta\tau$ as well the distribution of this statistic under the null.

Table 2: Causal Discovery over $G$ for various pairs of attributes, $(m_1, m_2)$. For each pair, the top row corresponds to taking $m_1$ as the treatment whilst the bottom utilizes $m_2$ as the treatment.

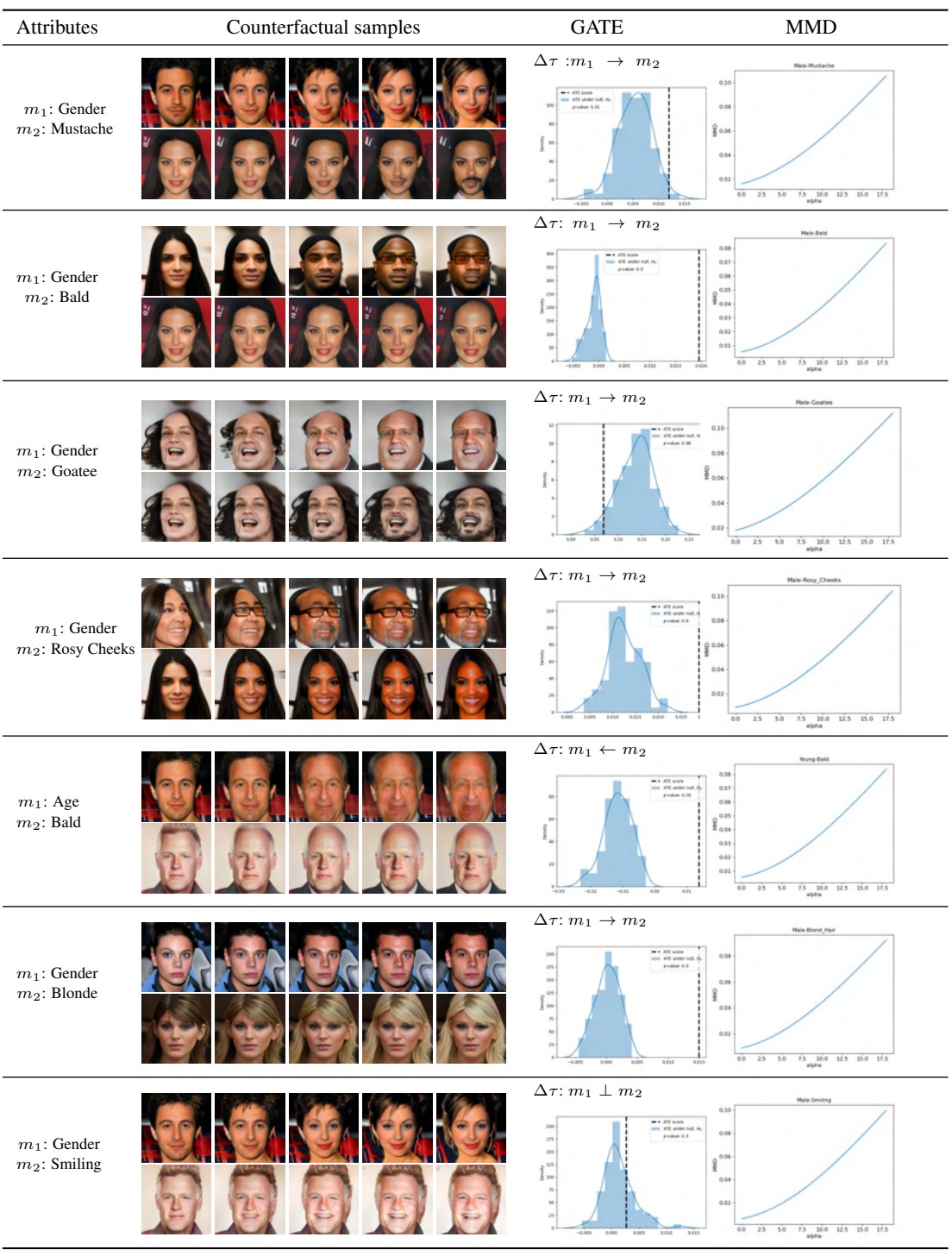

| Attributes | Counterfactual samples | GATE | MMD |
|---|---|---|---|
| $m_1$: Gender $m_2$: Mustache | | $\Delta\tau : m_1 \rightarrow m_2$ | |
| $m_1$: Gender $m_2$: Bald | | $\Delta\tau : m_1 \rightarrow m_2$ | |
| $m_1$: Gender $m_2$: Goatee | | $\Delta\tau : m_1 \rightarrow m_2$ | |
| $m_1$: Gender $m_2$: Rosy Cheeks | | $\Delta\tau : m_1 \rightarrow m_2$ | |
| $m_1$: Age $m_2$: Bald | | $\Delta\tau : m_1 \leftarrow m_2$ | |
| $m_1$: Gender $m_2$: Blonde | | $\Delta\tau : m_1 \rightarrow m_2$ | |
| $m_1$: Gender $m_2$: Smiling | | $\Delta\tau : m_1 \perp m_2$ | |

# B    BACKGROUND

## B.1    THE POTENTIAL OUTCOMES FRAMEWORK

We briefly overview the potential outcomes framework of Neyman (1923) and Rubin (1974), upon which we base our proposed method in the next section. In its simplest form, this framework considers the causal effect of assigning a binary treatment, $T \in \{0, 1\}$. Such a framework posits the existence of potential outcomes for the $i$th individual both whilst receiving treatment, $Y_{T=1}(i)$, and when treatment is withheld, $Y_{T=0}(i)$. The causal effect for a given individual, $i$, is defined as the difference in these potential outcomes.

The "fundamental problem of causal inference" is that whilst we may posit the existence of both potential outcomes, $Y_{T=1}(i)$ and $Y_{T=0}(i)$, we only ever observe the outcome under one treatment (Holland, 1986). For this reason, causal inference is often performed at the population level, by considering quantities such as the average treatment effect (ATE):

$$\tau = \mathbb{E}_i \left[ Y_{T=1}(i) - Y_{T=0}(i) \right]. \tag{4}$$

Equation (4) is the expected difference in potential outcomes of individuals receiving treatment $T = 1$ and $T = 0$. Under standard assumptions of unconfoundedness, positivity, and stable unit treatment value assumption, the ATE is identifiable and can be estimated by a statistical estimate of the associational difference in expectations (Rubin, 1978).

# C    MODEL DETAILS

For our experiments, whenever possible, we used the default settings found in the original papers of all chosen models. In particular, we used the default settings for both Mintnet (Song et al., 2019) and StyleGAN2 (Karras et al., 2020) which were pretrained on MNIST and FlickrFacesHQ respectively. For Mintnet we finetuned on a MorphoMnist dataset for 250 epochs using the Adam optimizer with default settings. Similarly, we also finetuned StyleGAN2 on CelebAHQ for  2000 iterations using 10% of CelebAHQ. Our synthetic experiments on the other required us to train a Masked AutoRegressive Flow (Papamakarios et al., 2017) that consisted of 10 layers with 4 blocks per layer. The Masked AutoRegressive Flow was trained for 5000 iterations using 5000 data samples.

For latent linear classifiers we use an SVM based classifier as found in the widely used Sci-kit learn library (Pedregosa et al., 2011). In Table 3 we report the test accuracy of our latent SVM based classifier and observe that many attributes of interest exhibit high degrees of linear separability. However, certain attributes such as Point Nose and Brown Hair are less linearly separable and as a result the causal relationships inferred by CAGE are less reliable.

| Attributes | SVM Classifier Accuracy |
|---|---|
| Gender | 90.0 |
| Mustache | 79.7 |
| Bald | 70.8 |
| Goatee | 83.2 |
| Rosy Cheeks | 72.8 |
| Age | 75.1 |
| Pointy Nose | 64.6 |
| Brown Hair | 65.4 |
| Smiling | 79.6 |
| Blond Hair | 84.8 |

Table 3: Latent Space SVM Classifier Test Accuracies on StyleGAN2 finetuned on CelebaHQ.

## C.1    HUMAN EVALUATION OF COUNTERFACTUALS ON AMAZON MECHANICAL TURK

To study the visual quality of generated counterfactuals by manipulating the cause or effect variable as determined by CAGE we solicit human evaluators on the Amazon mechanical turk platform. In

Figure 3: UI provided to AWS Turkers.

particular, we collect 100 generated pairs for each DAG in both the same source and same destination settings. Each pair of generated samples is then evaluated by 3 different human annotators giving a sum total of 300 annotations per DAG. A capture of the UI provided to the human annotators is depicted below 3.

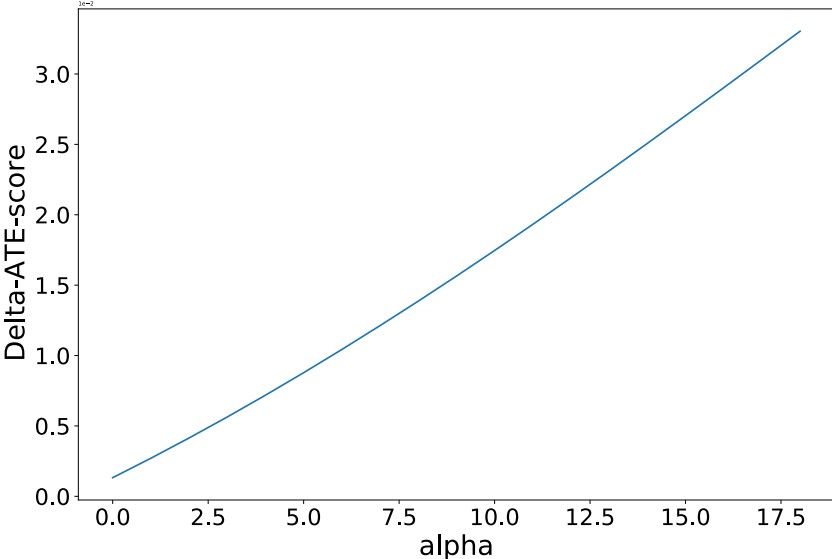

Figure 4: $\Delta\tau$ plot on the MorphoMNIST dataset as a function of $\alpha$.

## C.2 $\Delta\tau$ MorphoMNIST

We plot the $\Delta\tau$ as a function of interpolant strength below for the MorphoMNIST dataset. As observed $\Delta\tau > 0$ which indicates that under our generator $G$ thickness is the cause of intensity which matches the true synthetic data generation process.

### A NOTE ON BASELINE METHODS FOR CELEBHQ

We compared CAGE to well-established causal discovery algorithms when looking to infer structure over a deep generative model, $G$. However, the causal discovery baselines considered operate over the class probabilities output by probabilistic classifiers trained as §2.1.1. We note that while it would be possible to learn causal structure using baseline methods, such as LiNGAM, over the metadata (e.g., hair color and gender), this would not necesarly provide any insights into the causal structure implicit within the generator, $G$. For this reason, we instead focus on applying causal discovery methods over the output of proxy classifiers.

## D ADDITIONAL BASELINE EXPERIMENTS

In this section we perform causal discovery in the latent space using various popular approaches in the literature. As noted in the main paper these approaches cannot be applied to the observation space which are high dimensional images and are not baselines.

| Method | ANM | DAGs with No Tears | LinGAM |
|---|---|---|---|
| Gender, Mustache | $\longrightarrow$ | $\longrightarrow$ | $\longrightarrow$ |
| Gender, Bald | $\longrightarrow$ | $\longrightarrow$ | $\perp$ |
| Gender, Goatee | $\longleftarrow$ | $\longrightarrow$ | $\longleftarrow$ |
| Gender, Rosy Cheeks | $\longleftarrow$ | $\longrightarrow$ | $\longleftarrow$ |
| Age, Bald | $\perp$ | $\longrightarrow$ | $\perp$ |
| Pointy Nose, Brown Hair | $\longrightarrow$ | $\longrightarrow$ | $\perp$ |
| Gender, Smiling | $\longrightarrow$ | $\longrightarrow$ | $\perp$ |

**Hard vs. Soft**. As we have access to ground truth labels it is tempting to consider whether using hard labels as opposed to the classifiers probability is better suited to computing our $\Delta\tau$ metric. In the table below we repeat our causal discovery experiment over CelebAHQ. As observed, all baselines provide unreliable estimates to the causal relationship between variables when compared to the main table which is computed using soft labels. Finally, we found it useful to assign soft labels provided by the classifiers used in the ATE computations for all augmented images during group DRO training.

| Method | ANM | DAGs with No Tears | LinGAM |
|---|---|---|---|
| Gender, Mustache | $\perp$ | $\longrightarrow$ | $\perp$ |
| Gender, Bald | $\perp$ | $\perp$ | $\perp$ |
| Gender, Goatee | $\perp$ | $\perp$ | $\perp$ |
| Gender, Rosy Cheeks | $\perp$ | $\longleftarrow$ | $\perp$ |
| Age, Bald | $\perp$ | $\longleftarrow$ | $\longrightarrow$ |
| Pointy Nose, Brown Hair | $\perp$ | $\longleftarrow$ | $\longrightarrow$ |
| Gender, Smiling | $\perp$ | $\perp$ | $\perp$ |

**Baselines without using CounterFactuals**. We now turn to the use of Counterfactuals when computing our baseline scores. Specifically, we attempt causal discovery purely using observation data and labels with both hard and soft labels, which is in contrast to the main paper which considered baselines which had access to classifier probabilities on counterfactual data. The table below shows this result of this ablation.

| Method | ANM | DAGs with No Tears | LinGAM |
|---|---|---|---|
| Gender, Mustache | $\longrightarrow$ | $\longrightarrow$ | $\perp$ |
| Gender, Bald | $\longrightarrow$ | $\longrightarrow$ | $\perp$ |
| Gender, Goatee | $\longleftarrow$ | $\longrightarrow$ | $\perp$ |
| Gender, Rosy Cheeks | $\longleftarrow$ | $\longrightarrow$ | $\perp$ |
| Age, Bald | $\perp$ | $\longrightarrow$ | $\perp$ |
| Pointy Nose, Brown Hair | $\perp$ | $\longleftarrow$ | $\perp$ |
| Gender, Smiling | $\longrightarrow$ | $\longrightarrow$ | $\longrightarrow$ |

