# OpenReview forum: "Causal reasoning for controllable generative modeling "
_ICLR.cc/2022/Workshop/OSC — Submitted to ICLR2022 OSC _

### Official Review · Reviewer_RJTd · 2022-03-10
**Nice experiments, but flawed method**

**Rating:** 1
**Confidence:** 2

**Review:**

The authors propose a method to infer causal structure between some high-level variables using a pretrained generative model and a labeled dataset. To establish whether variable A causes variable B, they first train linear models to predict A and B from the latent space of the generative model. Next, they take samples from the data distribution, encode them to the latent space, and move them along the direction normal to the learned decision boundary for the variable A. They then say that a causal affect A -> B exists if the B model output changes during this latent traversal.

The problem setting certainly sounds interesting: it would be great if StyleGAN would learn a causal relation behind gender and baldness and we could infer that.

Unfortunately, I believe the approach to be fundamentally flawed. The proposed method will determine only if the latent direction corresponding to the feature A and the latent direction corresponding to the feature B are approximately orthogonal or not. Under some assumptions (that the authors could have been clearer about), this translates to an independence test between A and B. But if two features are statistically dependent, this method cannot possibly distinguish the cases of A causing B, B causing A, and A and B being confounded.

On the one hand, the authors seem to acknowledge this: they write that for this method to work, they require unconfoundedness and knowing the causal ordering of the variables a priori. Under these assumption, we can infer the existence of a causal link from independence tests, for instance with this method. (But we could achieve this easier, for instance by directly performing independence tests on the distribution of labels in the training data.)

On the other hand, the authors also suggest that we can infer the causal direction (A -> B or B -> A) by performing two such tests with flipped roles for A and B. I do not understand why this should work. If the true causal graph is A -> B, moving in the latent space along the B-associated direction does *not* correspond to an intervention on B, and in general may also change A. Take Figure 1, for instance: moving from z(i) vertically to the bottom (an "intervention" on e) will also cross the decision boundary for c. More generally, distinguishing the causal graphs A -> B and B -> A from purely observational data requires model assumptions, and it is not clear to me that this approach makes any assumptions that allow for causal discovery.

There are several aspects I like about the paper: the "counterfactual" samples from the experiments look good, the perceptual evaluation through human annotators is a nice touch, and the writing is generally good. Unfortunately, assuming I am not missing an important point, the core idea of the method is flawed. I therefore lean towards rejecting this paper.

Some minor points, questions, and typos:
- In the beginning, the paper advertises suitability for GANs, but later assumes the existence of an encoder without further comments.
- How is the intervention step size alpha chosen? How do the results depend on this choice?
- It's nice that the authors develop a procedure for estimating the null distribution of the test quantity. It is not clear to me though that the null hypothesis really makes sense, as with a reshuffling of the latent features we not only decorrelate the latent variables, but also remove any relation between the latent space and the variables, which then means the linear classifiers don't make sense any more.

---

### Official Review · Reviewer_yBoU · 2022-03-16
**Causal reasoning for controllable generative modeling**

**Rating:** 2
**Confidence:** 2

**Review:**

Overall: Paper is fairly clear, unsure regarding novelty, related work and citations referencing key assumptions borrowed from previous work are notably missing, but willing to give the authors the benefit of the doubt. Overall, assumptions that both the generative model and the classifier behave when encountered by unseen inputs, as is required by the proposal, are somewhat "swept under the rug". While the experiments section is sufficient, would suggest the authors invest effort in a more convincing evaluation demonstrating that the generative model and classifier do behave, and the proposed method does work and is applicable, in a wide variety of practically relevant scenarios.

Specifics:
- Why was terminology changed from $m_1$ and $m_2$ to $m_e$ and $m_c$?
- It should be noted that practically, the pretrained generative model will be imperfect, and thus there may be a distribution shift between the collected dataset and the generations from the perspective of the classifier.
- Furthermore, the do-intervention does not correspond to a sample from the latent prior, so the generation as well as the classifier in that case may perform less well than normal due to distribution shift.
- While the authors do address these points, I do take some issue with the following statement "modern deep generative models and classifiers can indeed satisfy the above requirements for many practical usecases (Denton et al., 2019; Brown et al., 2020)." There is plenty of evidence that classifiers generically are not calibrated, especially w.r.t. distribution shift. Furthermore, there is a body of literature, pushed by Eric Nalisnick and colleagues, on how generic generative models can behave poorly when fed unseen inputs. Simply citing the two works above while omitting the body of literature which highlights the brittleness of both generative models and classifiers is a fairly drastic omission.
- Citations and related work needs to be improved, i.e. statements like this "In addition to standard assumptions from causality (unconfoundedness, positivity, SUTVA)" are not very helpful at the very least without citations which readers can consult for the details.
- Grammatical checks, "We subsequently the generative average treatment effect (GATE)"

---

### Decision · Program_Chairs · 2022-03-21

**Decision:**

Reject

**Comment:**

Unfortunately the paper is not ready for presentation. I encourage the authors to take a closer look at the review from RjTd to revision their work towards a conference submission. The paper is interesting and reads well, I suggest the authors to focus on the following areas:
(1) the assumptions should be clarified
(2) the statistical test in latent space is very interesting but it needs more justification, also wrt the model being suboptimal
(3) the most controversial statement was around equation 3 and the assumptions for causal discovery. This should be clarified and made explicit.